# A New Exergy Disaggregation Approach for Complexity Reduction and Dissipative Equipment Isolation in Thermoeconomics

**DOI:** 10.3390/e24111672

**Published:** 2022-11-17

**Authors:** Rodrigo Guedes dos Santos, Atilio Barbosa Lourenço, Pedro Rosseto de Faria, Marcelo Aiolfi Barone, José Joaquim Conceição Soares Santos

**Affiliations:** 1Federal Institute of Espírito Santo—IFES, Vitória 29040-780, Brazil; 2Department of Mechanical Engineering, Federal University of Espírito Santo—UFES, Vitória 29075-910, Brazil; 3Federal Institute of Espírito Santo—IFES, Cariacica 29150-410, Brazil

**Keywords:** thermoeconomic approach, cost allocation, physical exergy disaggregation, modeling complexity, dissipative component

## Abstract

Thermoeconomics connects thermodynamic and economic concepts in order to provide information not available in conventional energy and economic analysis. Most thermoeconomicists agree that exergy is the most appropriate thermodynamic magnitude to associate with cost. In some applications, exergy disaggregation is required. Despite the improvement in result accuracy, the modeling complexity increases. In recent years, different exergy disaggregation approaches have been proposed, mostly to deal with dissipative components and residues, despite all of them also increasing the complexity of thermoeconomics. This study aims to present a new thermoeconomic approach based on exergy disaggregation, which is able to isolate dissipative components with less modeling complexity. This approach, called the A&F Model, splits the physical exergy into two terms, namely, Helmholtz energy and flow work. These terms were evaluated from a thermoeconomic point of view, through a cost allocation in an ideal Carnot cycle, and they were also applied and compared with the UFS Model, through a cost allocation analysis, in a case study with an organic Rankine cycle-powered vapor compression refrigeration system. The complexity and computational effort reduction in the A&F are significantly less than in the UFS Model. This alternative approach yields consistent results.

## 1. Introduction

For elaborate energy systems, the use of traditional energy analysis techniques—established on energy and mass balance (First Law of Thermodynamics)—to solve problems can be either hard, not enough, or inconclusive. Thus, thermoeconomics is a science that, by joining economics and thermodynamics, supplies knowledge to resolve these issues within energy cycles. For instance, logical cost estimation of cycle products is established on physical criteria [1].

The way in which we define the productive structure (fuel and product) is a key point of thermoeconomic modeling [2], and the most adequate thermodynamic magnitude to be associated with cost is exergy, according to several researchers [3,4]. Since it contains information from both the first and the second laws of thermodynamics, it qualifies the energy streams and identifies the irreversibilities of the subsystems.

Depending on the type of analysis, different levels of accuracy of the results are required, meaning that each thermoeconomic analysis requires a specific disaggregation level of the components and flows [4]. For local optimization design and operational diagnosis, for instance, total disaggregation of the system components is generally required in order to isolate each of the subsystems.

In agreement with Lozano & Valero [2], the deeper and more detailed the disaggregation is, the clearer the interpretation of the obtained costs will be and the wider the catalog of applications of theoretical and practical problems.

It is feasible to get more accurate results in thermoeconomic studies when the physical exergy is disaggregated. Nonetheless, there is a consequent increase in the modeling complexity [5,6].

The isolation of dissipative components and waste treatment represent two challenges in thermoeconomics. Waste treatment became more evident in the 1990s, during the CGAM problem evaluations [7], and it directly affects the cost allocation, making it difficult for the discipline to deal with environmental costs. In this regard, several studies have proposed and applied different methods for waste cost allocation in recent years [8,9,10]. The method introduced by [10] was extended into alternative methodologies by [8,9]. Recently, these concepts were used in an application of thermoeconomic diagnostics to a gas turbine [11].

The problem of dissipative component treatment (the isolation of condensers, valves, intercoolers, etc.) lies in the impossibility of defining their products and/or fuels using total exergy flows only. The total exergy flows cannot guarantee the explicit waste cost allocation (exhaust gases in gas cycle cogeneration plants and combined cycle systems). Thus, this thermoeconomic problem remains open [8], since the dissipative equipment cannot be isolated in the productive diagrams.

A usual solution is to merge the dissipative with productive equipment, such as condensers with turbines in steam cycles, valves and evaporators in refrigeration cycles, or intercoolers with compressors in gas cycles. However, this solution can be arbitrary, and it limits the quality of thermoeconomic analysis.

Frangopoulos [12] presented an elegant solution to isolate the condenser in the steam cycles, which used both total exergy and negentropy flows (E&S Model). Nonetheless, the negentropy is a fictitious flow [8,13]. Although this proposal allowed the fuel and product of the condensers to be defined, it has already been shown that this solution presents uncommon values of exergetic unit costs, i.e., lower than the unit [8,14]. Moreover, this solution does not allow us to define the fuel and product of the valves in order to isolate them in the productive diagrams.

In the study by d’Accadia & Rossi [15], thermoeconomic modeling of a refrigeration system was carried out, disaggregating both the valve and the condenser, and combining the concept of negentropy with the exergy disaggregated into its thermal and mechanical terms. Nevertheless, it maintains the uncommon results associated with the use of negentropy as a fictitious flow; and this type of disaggregation of exergy into thermal and mechanical terms [16] is only advisable, without arbitrariness, for fluids modeled as ideal gases [6]. Additionally, it cannot isolate condensers.

In order to achieve consistent values of exergetic unit costs, Santos et al. [17] approached the concept and magnitude of negentropy from a different perspective. The authors proposed it as an entropic term to be used in conjunction with enthalpy. In other words, the physical exergy was split into enthalpic (*H − H_0_*) and entropic components (*T*_0_*S* − *T*_0_*S*_0_). This model already considers the dissipative component (condensers and intercoolers) treatment and residue cost allocation naturally through its productive diagram, without the uncommon values obtained when the E&S Model is performed for cost allocation.

The H&S Model (*H: enthalpic term*, *S: entropic term*) is considered a consistent approach, but it is not capable of defining a product for any of the types of expansion or bypass valves. For the expansion valves of refrigeration and heat pump cycles, a solution proposed for the valve is the use of thermal and mechanical exergy disaggregation by performing the calculation methodology presented by [18,19]. However, it is not a generalized approach capable of isolating any type of valves. The UFS Model [20] (*U: internal energy term*, *F: flow work term*, *S: entropic term*) was then proposed as an extension of the H&S Model to solve this issue by disaggregating the physical exergy into three terms: internal energy term (*U* − *U*_0_), flow work term (*pV* − *p*_0_*V*_0_), and the entropic term (*T*_0_*S* − *T*_0_*S*_0_), while keeping all the characteristics of the H&S Model. Lourenço et al. [21] showed that internal energy, flow work, and entropic can be interpreted as physical exergy terms.

The UFS Model has been applied for cost allocation purposes [22], diagnosis applications [23], and combining cost allocation with environmental analyses [22,24].

At this point, it is worth emphasizing that exergy disaggregation for dissipative component isolation plays an important role, not only for cost allocation applications, but also for diagnosis and local design optimization through the Principle of Thermoeconomic Isolation [25].

One can say that the UFS Model is an extension of the H&S Model, and the application of the first one could only be justified if there were a valve in the system, because of the increase in modeling complexity. To be more precise, the UFS Model presents one more term of exergy to determine the product and fuel of all equipment in the productive diagrams in comparison with the H&S Model.

After presenting a review of the state of the art, it can be observed that a comprehensive methodology combined with a lower degree of associated complexity is always a point of interest and study in thermoeconomics studies. Bearing this in mind, this study aims to present a new exergy disaggregation approach to reduce the complexity of thermoeconomic modeling and also its computational efforts. This new approach, herein called the A&F Model (*A: Helmholtz energy term*, *F: flow work term*), disaggregates the exergy into only two terms: Helmholtz energy and flow work.

The novelty of this study is to present this new physical exergy disaggregation approach, the A&F Model, which adequately deals with dissipative components (condensers, intercoolers, and valves) with less modeling complexity.

It is evaluated from a thermodynamic and thermoeconomic point of view through cost allocation in an ideal Carnot Cycle, and it is also applied and compared with the UFS Model in an organic Rankine cycle-powered vapor compression refrigeration system.

## 2. Physical Exergy Disaggregation and Definitions of Fuel and Product

According to [13], physical exergy desegregation has been used in thermoeconomics since 1990 and was proposed by Tsatsaronis. In agreement with [6], the split of exergy improves the accuracy of the results.

The concept of the A&F Model begins with the disaggregation proposed by the UFS Model (internal energy, flow work, and entropic terms). It is possible to combine internal energy and entropic terms, such as *U* − *T*_0_*S*. This rearrangement has a thermodynamic meaning called Helmholtz energy [26]. Hence, the physical exergy disaggregation is proposed into Helmholtz energy combined with flow work (A&F Model) terms. Thus, the main advantage of this approach is the reduction of complexity associated with the productive diagram since the physical exergy disaggregation will be carried out in only two terms.

Moreover, there is no arbitrariness to disaggregating the physical exergy Equation (1) into its Helmholtz energy term Equation (5) and flow work Equation (6) terms when applying the A&F Model. Thus, for this disaggregation, the physical exergy of the i-th stream is given by Equation (1) when kinetic, potential, and other energy forms are neglected. From Equation (1), the terms of the H&S Model are obtained directly, without arbitrariness, according to the physical exergy of a flow.
(1)$${\dot{E}}_{i}^{PH}={\dot{E}}_{i}^{H}-{\dot{E}}_{i}^{S}={\dot{m}}_{i}\cdot \left[\left(h_{i}-h_{0}\right)-T_{0}\cdot \left(s_{i}-s_{0}\right)\right]$$

Applying the definition of specific enthalpy (*h = u + Pv*), Equation (2) is written.
(2)$${\dot{E}}_{i}^{PH}={\dot{m}}_{i}\cdot \left\{\left[\left(u_{i}+P_{i}\cdot v_{i}\right)-\left(u_{0}+P_{0}\cdot v_{0}\right)\right]-T_{0}\cdot \left(s_{i}-s_{0}\right)\right\}$$

Rearranging Equation (2), the three terms of UFS Model are obtained and given by Equation (3).
(3)$${\dot{E}}_{i}^{PH}={\dot{E}}_{i}^{U}+{\dot{E}}_{i}^{F}-{\dot{E}}_{i}^{S}={\dot{m}}_{i}\cdot \left[\left(u_{i}-u_{0}\right)+\left(P_{i}\cdot v_{i}-P_{0}\cdot v_{0}\right)-T_{0}\cdot \left(s_{i}-s_{0}\right)\right]$$

The first and third terms of Equation (3) are combined so that Equation (4) is written.
(4)$${\dot{E}}_{i}^{PH}={\dot{m}}_{i}\cdot \left\{\left[\left(u_{i}-T_{0}\cdot s_{i}\right)-\left(u_{0}-T_{0}\cdot s_{0}\right)\right]+\left(P_{i}\cdot v_{i}-P_{0}\cdot v_{0}\right)\right\}$$

The specific Helmholtz energy of a closed system under a heat bath (reservoir at *T*_0_) is given by *a = u* − *T*_0_*s*. This can be applied to both *i*-th and dead states. Equations (5) and (6) show the Helmholtz energy term and flow work terms, respectively. In this way, the physical exergy could be written as in Equation (7), which represents a novel exergy disaggregation model.
(5)$${\dot{E}}_{i}^{A}={\dot{m}}_{i}\cdot \left(a_{i}-a_{0}\right)={\dot{m}}_{i}\cdot \left[\left(u_{i}-T_{0}\cdot s_{i}\right)-\left(u_{0}-T_{0}\cdot s_{0}\right)\right]$$
(6)$${\dot{E}}_{F}^{i}={\dot{m}}_{i}\cdot \left(P_{i}\cdot v_{i}-P_{0}\cdot v_{0}\right)$$
(7)$${\dot{E}}_{i}^{PH}={\dot{E}}_{i}^{A}+{\dot{E}}_{i}^{F}$$

In this study, the novel approach shall be called the A&F Model. It is worth mentioning that, for the thermoeconomic methodologies, one of the main points is to know the cost formation process and identify the causes that generate the cost in order to define a function or a productive purpose for each subsystem and plant [13]. It is noteworthy that different productive structures generate different costs [2]. According to [4], irreversibility is the physical magnitude generating the cost. Thus, irreversibility is, qualitatively, the cause of cost generation. Efficiency is the proportion of the desired result for an event to the inlet required to achieve it [27]. Moran et al. [26] stated that efficiency gauges how effectively the input is converted to the product. Hence, the concepts of fuel (*Fu*), product (*Pr*), irreversibility (*Ir*), losses (*Lo*), and efficiency (*ƞ*) are not independent in thermoeconomics, as evidenced in Equation (8) and (9). Equation (8) describes a general balance for each unit and the whole plant. The resources (fuel, *Fu*) expended are not fully converted into the product (*Pr*) or service due to the irreversibilities (*Ir*) and losses (*Lo*) associated with the production process. In other words, the efficiency defined by the product fuel ratio is, quantitatively, the parameter for the cost generation index.
(8)Fu-Pr=Ir+Lo
(9)$$\eta =\frac{Pr}{Fu}$$

It is worth mentioning that in cost accounting methodologies, the values of costs strongly depend on the *Pr/Fu* definition of the devices and on the set of auxiliary equations. In other words, the values of costs change depending on the *Pr/Fu* definition in the subsystems and also depending on how the different subsystems are interconnected in the productive structure [4].

In the A&F Model, the physical exergy is disaggregated into Helmholtz energy and flow work terms. The products and the fuels of all subsystems, in terms of Helmholtz energy and flow work terms, are determined in accordance with the quantity of these magnitudes added to and removed from the working fluid, respectively. These two new terms have a positive contribution to the physical exergy (Equations (4) and (7)), and the consideration allows one to define the fuels and the products of the dissipative and productive component.

## 3. Thermoeconomic Modeling

The thermoeconomic model must be accomplished by utilizing Equations (10) and (11). The resolution of this cost equation set, Equation (10), is the monetary unit costs of all internal flow and final product. The monetary unit cost of a flow is the amount of external monetary units required to obtain one unit of this flow, meaning that the monetary unit costs of a flow are the measure of the economic efficiency of the production process when producing it [4].
(10)$$\sum \left(c_{out}\cdot Y_{out}\right)-\sum \left(c_{in}\cdot Y_{in}\right)=c_{F}\cdot E_{F}+Z$$

In Equation (10), *Z* represents the external hourly cost of the subsystem due to the capital, operation, and maintenance costs of each subsystem (in $/h); $E_{F}$ is consumption (kW) and $c_{F}$ is the monetary unit cost, both from the external fuel exergy. The unknown variables *c_out_* and *c_in_* are the monetary unit cost of the internal flows at the outlet and at the inlet of each subsystem (in $/kWh), respectively; and $Y_{out}$ and $Y_{in}$ mean the generic thermodynamic magnitude of the internal flows at inlet and outlet of each subsystem. The solution of the set of equations results in the monetary unit costs of each internal flow and each final product [28]. In this paper, Y assumes the thermodynamic magnitudes, for power (*W*), heat exergy (*Q*), total exergy (*E*), Helmholtz energy term (*E^A^*), flow work term (*E^F^*), internal energy (*E^U^*), and entropic term (*E^S^*).

In Equation (11), the unknown $k_{out}$ and $k_{in}$ are the exergetic unit costs of the internal flows (output/input) of every subsystem; and the hourly cost of the subsystem, due to the capital cost, operation, and maintenance, must be zero (*Z* = 0). Since there is no information about the resource, $k_{F}$ is assumed to be 1.00 kW/kW [4]. The exergetic unit cost of a flow is the amount of external exergy unit required to obtain one unit of this flow, meaning that the exergetic unit cost of a flow is a measure of the thermodynamic efficiency of the production process when producing this flow [4].
(11)$$\sum \left(k_{out}\cdot Y_{out}\right)-\sum \left(k_{in}\cdot Y_{in}\right)=k_{F}\cdot E_{F}$$

Each subsystem has a single cost equation, thus auxiliary equations are necessary. Thermoeconomic models based on the productive diagrams consider the equality criteria [29,30,31], where productive flows exiting the same productive unit must have the same unit cost.

## 4. Case Studies

The Carnot refrigeration cycle and a system combining the organic Rankine cycle (ORC) and vapor compression refrigeration (VCR) cycle, subsequently called the ORC–VCR system, are evaluated from a thermodynamic and thermoeconomic point of view through cost allocation using the A&F Model. Moreover, by using these plants, this study shows the capacity of the A&F Model to treat both kinds of dissipative components in thermoeconomics: valves and condensers.

### 4.1. Carnot Refrigeration Cycle

The beauty of a theory is usually shown in the simplicity of its forms and the generality of its message, but its power resides in its capacity to solve practical cases [32]. Bearing this in mind, for the purpose of illustrating the application of the A&F Model, one simple example is used to evaluate the consistency and coherence of the model from both the thermodynamic and thermoeconomic point of view: the Carnot refrigeration cycle.

There is no refrigeration cycle that operates between two specified temperature levels (in this case, 0 and 26 °C) that is more efficient than the theoretical Carnot refrigeration cycle. The compressor (cmp), the evaporator (evp), the turbine (trb), and the condenser (cnd) are the four units (or subsystems) that compound the physical structure presented in Figure 1.

Table 1 shows the principal physical flows of the Carnot refrigeration cycle studied and its principal parameters. The mechanical power demanded by the compressor is 13.8 kW. Freon R-134a is the refrigerant used in this cycle, and its mass flow is 0.8 kg/s. The thermodynamic properties of R-134a are evaluated from the database of the software Engineering Equation Solver [33].

Figure 2 illustrates the productive diagram for the Carnot refrigeration cycle using the A&F Model. The variation of the Helmholtz energy term (*E^A^_l:m_*) and flow work term (*E^F^_l:m_*) components of the exergy between two physical states, l and m, is used to define the flows. In Figure 2, the rectangles are the real units (or subsystems) that represent the actual equipment of the system. The hexagon and the circles are fictitious units, the purpose of which is to join and/or bifurcate the productive flows. Every subsystem in Figure 2 has input arrows that symbolize its fuel (or resource) and output arrows that symbolize its products. The definition of productive flows is attained from the specific exergy term variation between the inlet and outlet. For example, it is classified as a product when this variation is positive and the opposite (negative variation) as a fuel [17]. Multiproduct is the method used to formulate the auxiliary equation in each bifurcation. This method assumes that all productive flows exiting the same subsystem or unit have the same unit cost since they are exiting from the same subsystem with the same resources and irreversibilities [34].

### 4.2. ORC–VCR System

The physical structure of the ORC–VCR system is shown in Figure 3. This system consists of a combination of two cycles: the ORC, identified as 1–2–3–4–1; and the VCR cycle, as 5–6–3–7–5 [35]. The ORC and VCR cycles utilize the R-600 as a working fluid. The features are that the shafts of the expander of the ORC and the compressor of the VCR cycle are directly coupled [35]. The mechanical power delivered from the ORC, through the expander, is enough to drive the compressor, the boiler feed pump [35], and to produce 10 kW of net power. The ORC and VCR cycles share a common condenser [35].

Table 2 presents the main parameters of the principal physical flows of the ORC–VCR system at design point thermodynamic modeling. The refrigerant is hydrocarbon R-600, of which the mass flow is 1.00 kg/s for the ORC cycle and 0.362 kg/s for the VCR cycle. The thermodynamic properties of R-600 are evaluated from the database of the software Engineering Equation Solver [33]. For the defined design point simulation, the isentropic efficiencies of the expander, the compressor, and the boiler feed pump are 0.80, 0.75, and 0.75, respectively [36].

Table 3 shows the formula for the product-fuel ratios of each equipment present in the physical structure and their respective values according to three thermoeconomic models: E, A&F, and UFS.

It is relevant to note that the E model is incapable of defining, without some degree of arbitrariness, the product of the condenser and valve (dissipative equipment). Thus, for these kinds of equipment, the exergy variation is defined as fuel, and it would not be possible to quantify or define the condenser and valve products (or function) in terms of total exergy. Therefore, they could not be isolated in the productive diagram if the E model were considered for thermoeconomics analysis. As an alternative, this kind of equipment could be aggregated with any other component. The authors must then decide, arbitrarily, whether to merge in order to arbitrate its productive purpose. Since this is not the purpose of this research, this decision to merge the dissipative equipment is not performed here. However, the E model was chosen to compare the results of the remaining models (A&F and UFS) from a thermodynamic point of view only, avoiding unnecessary arbitrariness.

Figure 4 shows the productive diagram for the ORC–VCR system utilizing the UFS model.

Figure 5 shows the A&F Model being used in an ORC–VCR system and represented in a productive diagram. When the input and the output of each subsystem are defined by the Helmholtz energy term (*E^A^*) and flow work (*E^F^*) terms, all equipment in the cycle is isolated, including the valve and condenser (dissipative equipment).

By comparing Figure 4 and Figure 5, it is observed that the A&F model has reduced the quantity of flows by approximately 27%, the quantity of junctions–bifurcations by 33%, and the quantity of cost equations by 38%, in relation to the UFS Model. Thus, the complexity in the A&F is significantly smaller than in the UFS Model.

## 5. Results and Discussions

The thermoeconomic results of the case studies presented in Section 4 are presented in Section 5. The exergetic unit cost of the Carnot refrigeration cycle by the A&F Model in Section 5.1 and a comparison discussion between the A&F Model and the UFS Model, considering both the design point analysis in Section 5.2 and a parametric study analysis, is presented in Section 5.3 for the ORC–VCR.

### 5.1. Carnot Refrigeration Cycle

It is possible to observe, when using the A&F Model in the productive diagram for the Carnot refrigeration cycle, the values of the productive flows and their respective exergetic unit costs in Table 4.

When applying the cost allocation of a thermoeconomic model in a Carnot refrigeration cycle, it is expected (from a thermodynamic point of view) that the value of the products and the fuels of each subsystem are equal and, consequently, the exergetic unit cost of each productive flow and the efficiency are equal to one (there are no irreversibilities). Additionally, the four conversion processes of this cycle are isolated in the productive diagram. All of this information can be observed and comprised in Table 4 and/or Figure 2. It may, therefore, be affirmed that the A&F Model is a consistent approach.

### 5.2. Design Point (ORC-VCR)

Table 5 shows the values of fuel (*Fu*), product (*Pr*), irreversibility (*Ir*), and efficiency (*η*) of each component of the plant according to the approaches applied. Note that although they define different productive structures (different fuel and products in each production unit, as can be seen in Figure 4 and Figure 5), the irreversibility (the entity generating costs) of each component is the same. However, the methodologies determine different costs since the fuels and products of the productive units are different. It is extremely important to point out that different productive structures (fuel and product) generate different costs [2] and, depending on the productive structure definition (fuel and product), different cost values can be obtained [4]. By comparing the E Model with the A&F Model, three pieces of equipment have the same efficiency (boiler, feed pump, and compressor). It is plausible since the sum of Helmholtz energy and the flow work defines the total physical exergy, and the Helmholtz energy is a summation of internal energy and entropic terms.

It is worth mentioning that the A&F Model obtains efficiency values equal to or closer to the conventional efficiency based on total exergy flow (E Model) than the UFS Model. In addition to isolating the valve and condenser with a coherent product/fuel ratio, the A&F Model does not significantly differentiate the efficiency of other productive equipment concerning the E Model.

By using the A&F Model, the product-fuel ratios (efficiency) have results between zero (if the processes are totally irreversible) and one (if the processes are totally reversible) for each component present in the plant (even the dissipative ones, such as the condenser), as can be seen in the results of this study presented in Table 5. The same conclusion can be extracted for the product-fuel ratio considering the UFS model.

Finally, note in Table 5 that although the models define different productive structures (different fuel and products in each production unit), the irreversibility (*Ir*) (the entity generating costs) of each component is the same. Thus, it can be stated that the models are coherent from a thermodynamic point of view. The cost allocation models had the challenge of determining the exergetic unit cost of net power and useful heat. Regardless of the allocation method, the results are a pair of exergetic unit costs for both final products along the defined straight line. Thus, for each model, the higher the exergetic unit cost of the useful heat exergy, the lower the exergetic unit cost of the net power. The straight-line solution is formed by two thermodynamic parameters: the overall exergetic efficiency and the ratio between the final products of the plant. Thus, independently of the kind of cogeneration plant, the straight line will be the same if there is no change at these two parameters. Figure 6 illustrates the straight line with the results estimated by the A&F and UFS Models for the case study. The useful heat exergy and net power unit costs can be considered as coherent results when intercession of both (ordered pairs) are on the line with the solution, as can be observed at the two points in Figure 6. The comparison between both methodologies, using the productive diagram, shows a distinction in the exergetic unit cost of net power of around 17%. Consequently, the useful heat unit cost differs by around 20% between both models.

Table 6 confirms that when comparing the A&F and UFS thermoeconomic models, a distinct behavior can be observed, meaning that the A&F model overcharges the useful heat to the detriment of net power. This is explained by the difference in the thermodynamic properties associated with the product and the fuel for each subsystem.

Figure 7 illustrates the results obtained for the subsystem product unit cost by the A&F and UFS models. Even when a device has more than one product, the exergetic unit costs of these products are the same. There is a similar tendency in the cost behavior of equipment products. For example, boiler products have the lowest exergetic unit cost; and this occurs because it is where the system cost formation process begins. The valve, on the other hand, has the highest exergetic unit cost for products among all equipment. It is noteworthy that all products in the system have a unit cost greater than one.

Although the exergetic efficiency of the boiler in the UFS model has a high value, the entropic term, that is, fuel for this equipment, has a high cost. Consequently, the cost of the products for this equipment by the UFS Model is also high. Regarding the A&F model, which has the same efficiency as the E model, despite not being highly efficient, its fuel is only associated with fuel cost, therefore the costs of its products are not high.

Table 7 shows all the productive flows, their internal energy, flow work, entropic, and Helmholtz terms values, and their respective exergetic unit costs. It is noteworthy that all the results are consistent since they are not less than one.

For all the models used in this study, the products of the equipment are less than its fuels. Consequently, all costs are greater than one.

Regarding the modeling complexity, by comparing the A&F and UFS models represented in the productive diagrams for the ORC–VCR system (Figure 4 and Figure 5), it is observed that the A&F model has reduced the quantity of flows by approximately 27%, the quantity of junctions–bifurcations by 33%, and the quantity of cost equations by 38%, in relation to the UFS Model. Thus, both models are able to isolate the dissipative components of the system, but the complexity associated with the A&F model is significantly smaller than that of the UFS model. This comparison of complexity considering the quantities of flows, junctions–bifurcations, and auxiliary equations is highlighted in Table 8.

### 5.3. Parametric Study (ORC–VCR)

A parametric analysis is carried out to investigate the effects of several parameters, such as equipment efficiency and temperature variations, on the exergetic unit cost and product-fuel ratio (*P/F*), obtained by applying the E, A&F, and UFS models in the case study of the ORC–VCR system. The E model was applied for thermodynamic reference only, avoiding unnecessary arbitrariness involved in dissipative equipment.

Table 9 shows the variation of expander isentropic efficiency and its influence on the product-fuel ratio according to the E, A&F, and UFS Models. Figure 8 shows how the exergetic unit cost of the expander product and product-fuel ratio (*P/F*) vary with an expander isentropic efficiency variation. The higher the isentropic efficiency, the higher the exergetic efficiency (*P/F*) due to the lower irreversibility generation for both the A&F and UFS models. In addition, the *P/F*, regardless of the thermoeconomic model, increases until it reaches 1 for a fully reversible and adiabatic process (efficiency of 100%). Consequently, the exergetic unit cost will be lower, as shown in Figure 8. Although the *P/F* ratio seems to be the same (overlapping) in Figure 8, there is a small difference between the E model and the remaining ones (A&F and UFS), as shown in Table 9. This small difference was expected due to the different manner that the E model defines the product and fuel of the expander compared to the A&F and UFS models (Table 3).

A similar analysis is performed for the pump and the compressor. The results for the product-fuel ratio and exergetic unit cost present the same tendency as those obtained in the analysis of the expander, increasing the isentropic efficiency and reducing the unit cost, due to the increase in the *P/F*. The same kinds of results are obtained for the compressor. For the pump, the results are shown in Table 10 and Figure 9. For the compressor, similar results are presented in Table 11 and Figure 10. In these analyses, for the three mentioned components (expander, pump, and compressor), the variation in the exergetic unit cost of the equipment products, obtained by the A&F and UFS models, is in accordance with the definition of the fuel and product of each model in its productive diagram, as shown in Table 3 and explained in Section 4.2.

For the evaporator and boiler, the parametric analysis is carried out by increasing the temperature variation (Δ*T*) between the working fluid and external reservoirs, T_COLD_ and T_HOT_, respectively. Regarding the exergetic unit cost, both analyses for the evaporator and boiler present the same tendency: the increase in the temperature difference (Δ*T*) causes more irreversibilities associated with heat exchange, and therefore the unit cost increases. Nevertheless, the A&F model obtains unit costs higher than the UFS model for the evaporator (Figure 11), while the opposite happens for the boiler (Figure 12). This is due to the different behavior in the *P/F* definition (Table 3) since the working fluid has a fixed temperature, the component with greater fuel will present greater irreversibility and, therefore, greater unit cost.

For both the evaporator and the boiler, the increase in the temperature difference (Δ*T*) between the reservoirs and the working fluid reduces the *P/F*, as shown in Table 12 and Figure 11 for the evaporator, and Table 13 and Figure 12 for the boiler. In both cases, there is an increase in the fuel for a constant product, so the product-fuel ratio decreases and, consequently, the unit cost increases.

From a thermodynamic point of view, the A&F model has advantages over the two other models applied in this work. In relation to the E model, it allows isolating dissipative equipment and still presents product-fuel ratio values that are very close or equal, which is different from the UFS model. For example, for the compressor and expander, respectively, in Figure 13a,b, the efficiency values for the E and A&F models are practically the same, which does not happen in the UFS model. It is important to highlight that they are all consistent since for isentropic efficiency equal to 1, all have an efficiency value equal to 1.

In the case of heat exchangers, the same tendency is observed for the boiler and evaporator, respectively, in Figure 14a,b. The values of the product and fuel ratio between Models A&F and E are very close. It is necessary to emphasize that, in this case, the values are also coherent because, for a temperature variation equal to zero, the efficiency value of the evaporator models is equal to 100%. In relation to the boiler, this does not happen because the addition of heat does not happen at a constant temperature; therefore, it is not an internally reversible process.

All these analyses, considering the design point and the parametric study, allowed us to conform to the proposed objective of this study, allowing us to extract the conclusion below.

## 6. Conclusions

This work presented a new exergy disaggregation approach, herein called the A&F model, which is a solid option for total disaggregation of each subsystem in a thermodynamic cycle, especially when there are dissipative components under analysis. The physical exergy terms of this model have a thermodynamic meaning: Helmholtz energy and flow work. The results of application to a Carnot refrigeration cycle confirmed consistent values of exergetic unit costs and productive efficiencies were yielded, so that the new approach agrees with the Second Law of Thermodynamics. The product-fuel ratios must have their results between zero (when the processes are totally irreversible) and one (when the processes are totally reversible) for each component present in the plant, as can be seen in the results of this study.

The complexity in the A&F model can be considered much lower than in the UFS model due to the decrease in the number of exergy terms and, consequently, in the productive flows and junctions/bifurcations encompassed in the analysis.

Parametric analyses were carried out in order to investigate the effects of some parameters on the exergetic unit cost and on the product-fuel ratio. The results confirm the consistency of the A&F model, as they present the expected tendency. By increasing the temperature variation (∆*T*) between the working fluid and external reservoirs, the cost has to increase and the P/F ratio has to be reduced. On the other hand, the efficiency increase reduces the exergetic costs due to lower irreversibility generation; and, the *P/F*, regardless of the thermoeconomic model, increases until it reaches 1 for a fully reversible process and adiabatic (efficiency of 100%). It is exactly what happens with the A&F and UFS models.

It is worth mentioning that the A&F model obtains efficiency values equal to or closer to the conventional efficiency based on total exergy flow (E model) than the UFS model. In addition to isolating the valve and condenser with a coherent product-fuel ratio, the A&F model does not significantly differentiate the efficiency of other production equipment concerning the E model. Even though defining different productive structures (fuel and product), the magnitude of generating costs (irreversibilities) is the same in all components of the plant for all models. Therefore, the different thermoeconomic methods achieve different exergetic unit costs.

The A&F model is an alternative for physical exergy disaggregation, keeping the vast majority of characteristics as the UFS model. The work was proposed to present a new approach to physical exergy disaggregation as a solid and strong option to resolve hard questions in thermoeconomics associated with the treatment and isolation of dissipative equipment and to reduce the complexity in thermoeconomic modeling since the physical exergy disaggregation will be carried out in two terms only. By comparing the productive diagrams of both models, the A&F model has reduced the quantity of flows by approximately 27%, the quantity of junctions–bifurcations by 33%, and the quantity of cost equations by 38%, in relation to the UFS model. Thus, the complexity and computational effort reduction in the A&F model are significantly smaller than in the UFS model.

Furthermore, this novel approach can be easily applied in different refrigeration systems, especially when there are dissipative components in the cycle under analysis, because it is a strong option for disaggregation of each subsystem in thermodynamic plants. In future works, the A&F model can be applied to other thermodynamic systems, and based on the results so far, the expectations are positive.

## Figures and Tables

**Figure 1 entropy-24-01672-f001:**
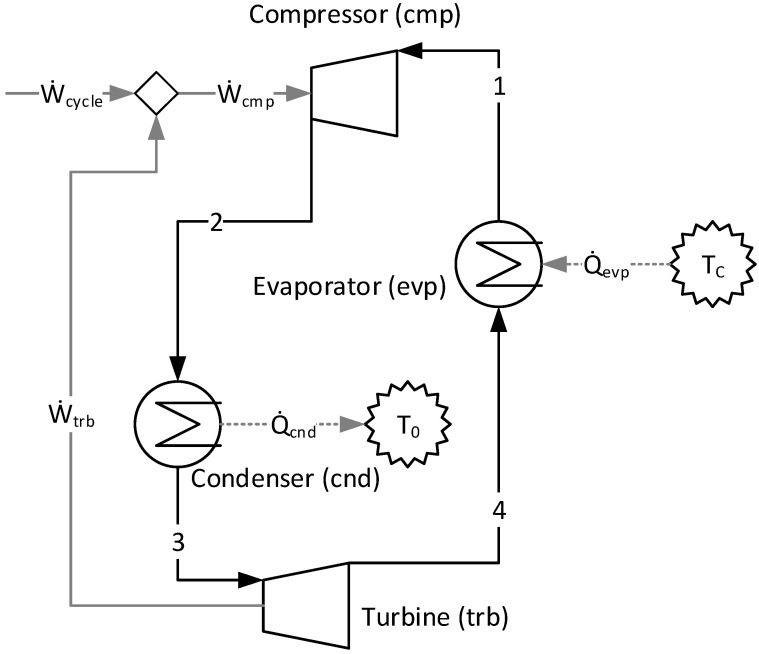
The physical structure of Carnot refrigeration cycle.

**Figure 2 entropy-24-01672-f002:**
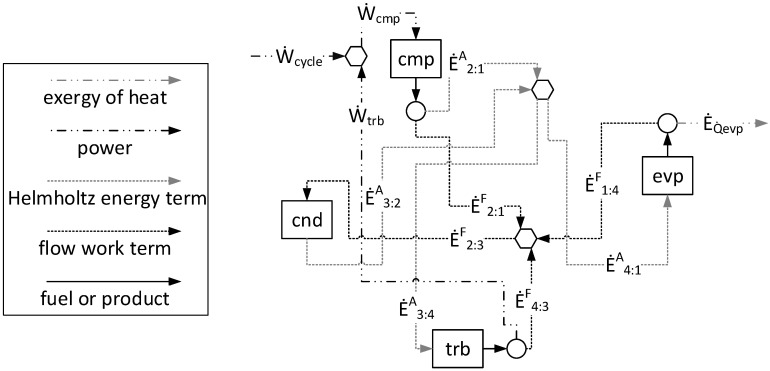
The A&F Model represented in a productive diagram for the Carnot refrigeration cycle.

**Figure 3 entropy-24-01672-f003:**
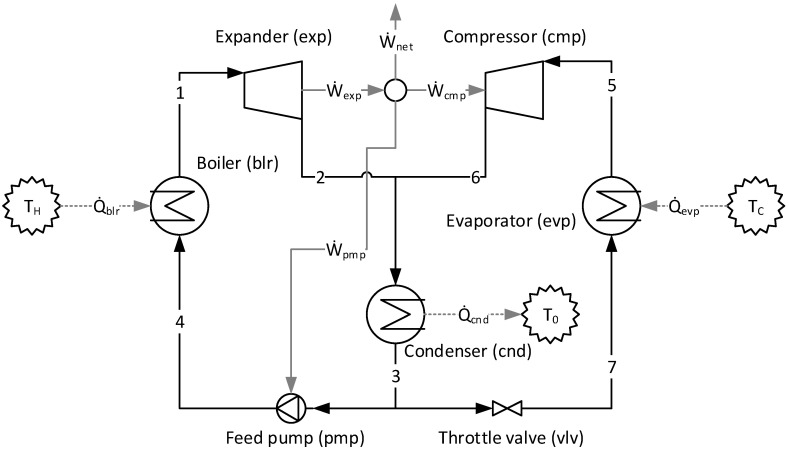
The physical structure of the ORC–VCR system (adapted from Li et al., 2013).

**Figure 4 entropy-24-01672-f004:**
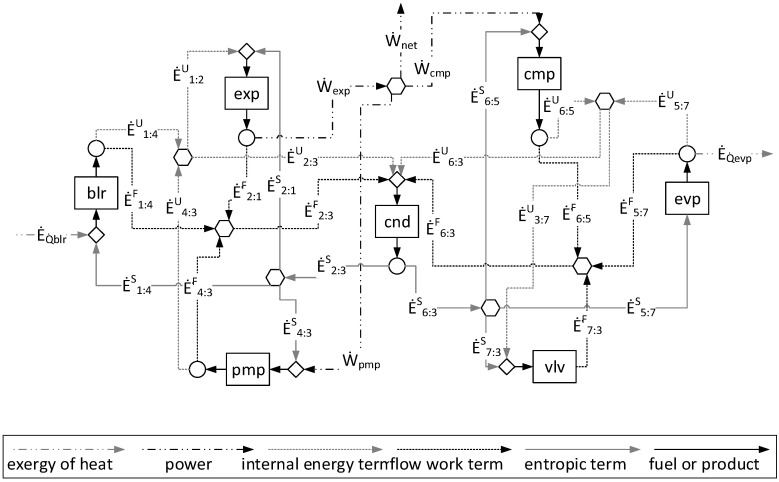
The UFS model represented in a productive diagram for the ORC–VCR system.

**Figure 5 entropy-24-01672-f005:**
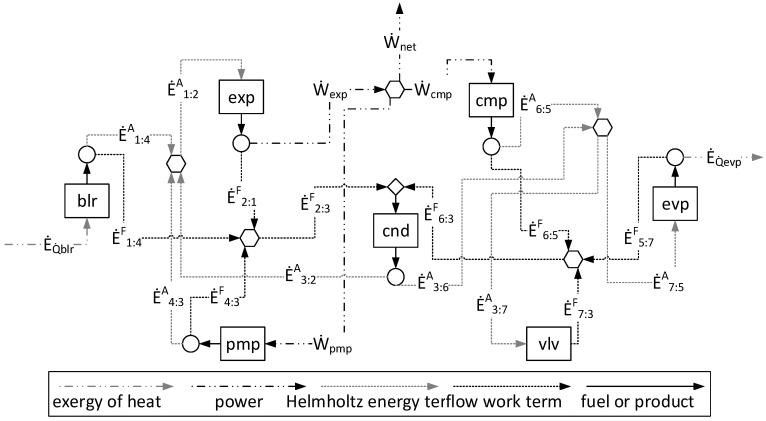
The A&F Model represented in a productive diagram for the ORC–VCR system.

**Figure 6 entropy-24-01672-f006:**
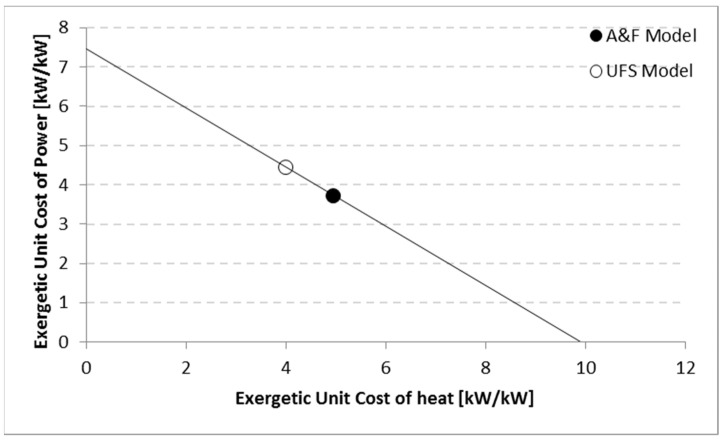
Results of both final products unit costs obtained by the A&F and UFS Models.

**Figure 7 entropy-24-01672-f007:**
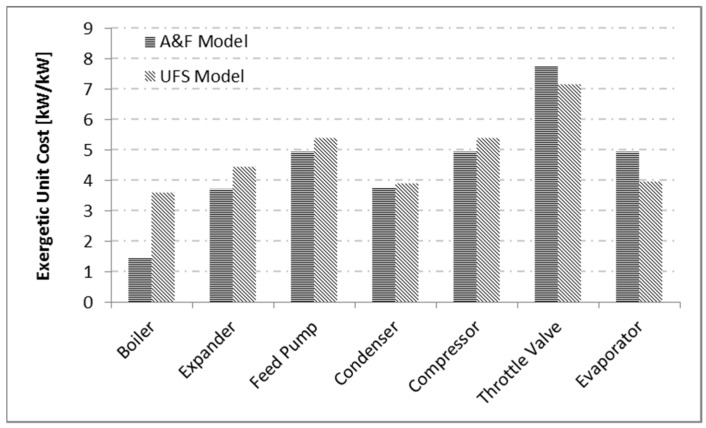
Results obtained for the subsystem product unit cost by the A&F and UFS Models.

**Figure 8 entropy-24-01672-f008:**
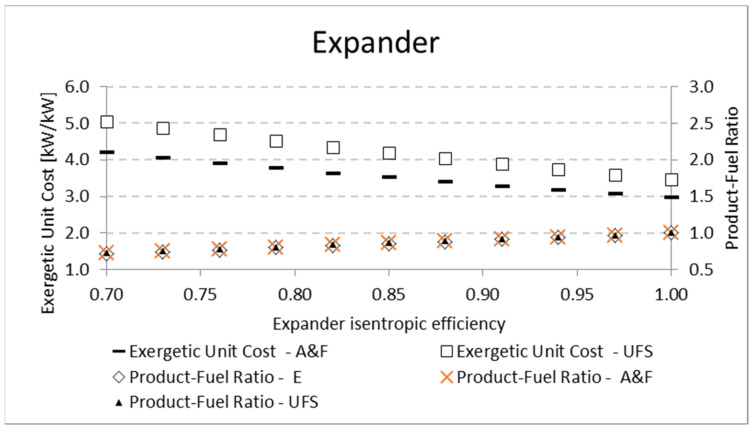
Exergetic unit cost and product-fuel ratio variations of the expander due to its isentropic efficiency.

**Figure 9 entropy-24-01672-f009:**
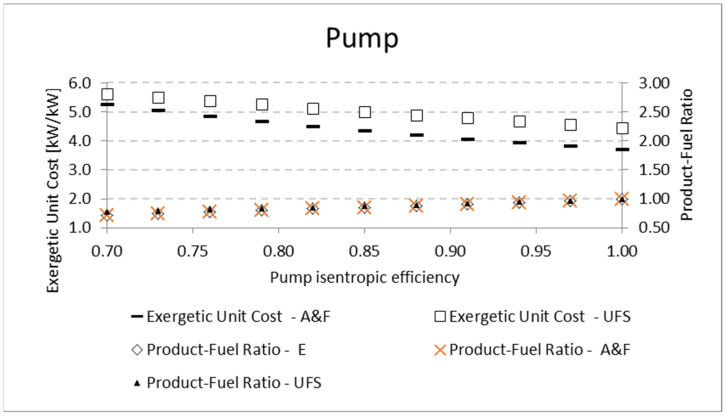
Exergetic unit cost and product-fuel ratio variations of the pump due to its isentropic efficiency.

**Figure 10 entropy-24-01672-f010:**
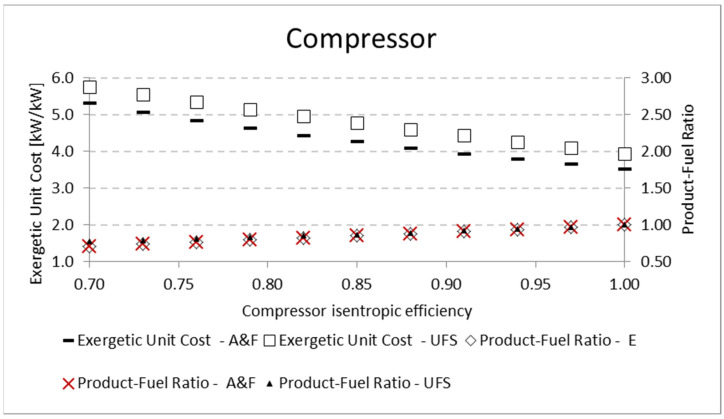
Exergetic unit cost and product-fuel ratio variations of the compressor due to its isentropic efficiency.

**Figure 11 entropy-24-01672-f011:**
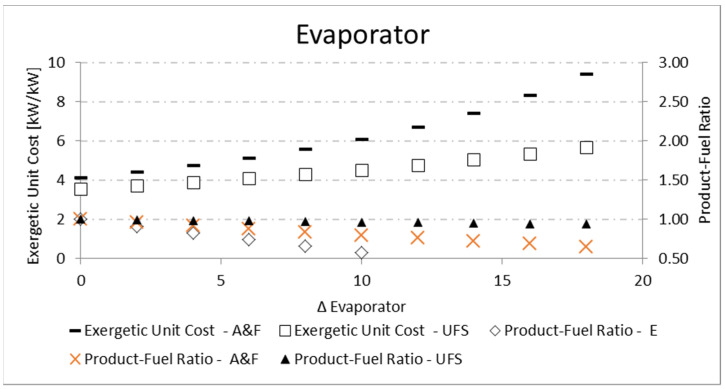
Exergetic unit cost and product-fuel ratio variations of the evaporator due to its (Δ*T*).

**Figure 12 entropy-24-01672-f012:**
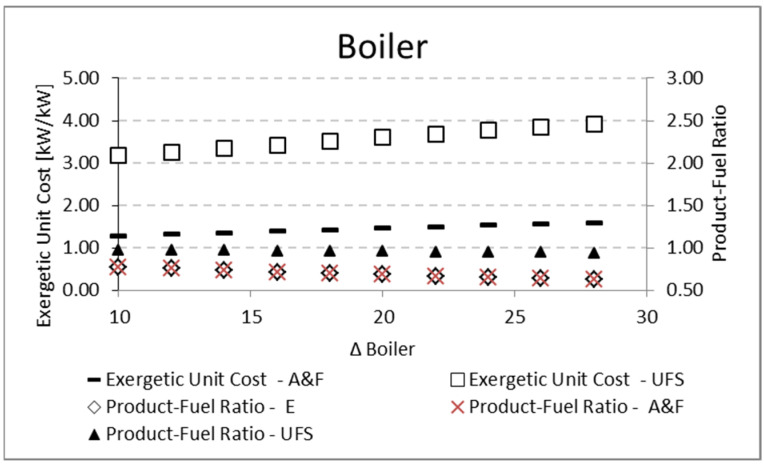
Exergetic unit cost and product/fuel ratio variations of the boiler due to its (∆*T*).

**Figure 13 entropy-24-01672-f013:**
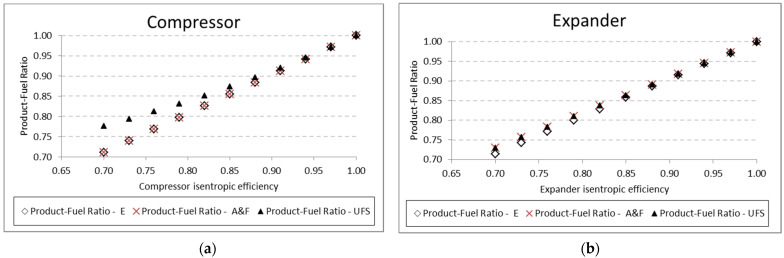
Product-fuel ratio variations due to its isentropic efficiency for: (**a**) compressor and (**b**) expander.

**Figure 14 entropy-24-01672-f014:**
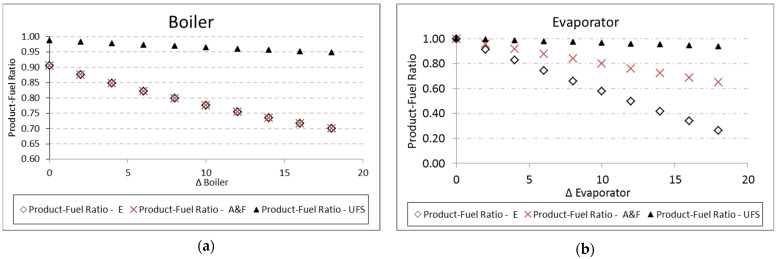
Product-fuel ratio variations due to its (∆*T*) for: (**a**) boiler and (**b**) evaporator.

**Table 1 entropy-24-01672-t001:** Principal parameters of the principal physical flows of the Carnot refrigeration cycle.

Physical Flow		*M* [kg/s]	*p* [kPa]	*T* [°C]
N°	Description			
1	Mixture (*x* = 0.985)	0.8	293	0
2	Vapor (*x* = 1)	0.8	685.8	26
3	Liquid (*x* = 0)	0.8	685.8	26
4	Mixture (*x* = 0.171)	0.8	293	0

**Table 2 entropy-24-01672-t002:** Principal parameters of the physical flows of the ORC–VCR system.

Physical Flow		m [kg/s]	p [kPa]	T [°C]	v [m3/kg]	e^U^ [kJ/kg]	e^H^ [kJ/kg]	e^S^ [kJ/kgK]	e^A^ [kJ/kg]	e^F^ [kJ/kg]
N°	Description									
1	Vapor	1.00	1013.00	80	0.040	656.10	696.30	2.47	−94.02	40.26
2		1.00	379.20		0.112	621.60	663.90	2.50	−136.10	42.28
3	Liquid	0.36	379.20	40	0.002	295.90	296.60	1.33	−106.75	0.68
4		1.00	1013.00		0.002	296.30	298.10	1.33	−106.65	1.82
5	Vapor	0.36	124.40	5	0.305	553.80	591.60	2.41	−176.31	37.88
6		0.36	379.20		0.108	608.40	649.40	2.45	−135.65	40.92
7		0.36	124.40		0.069	287.90	296.60	1.35	−120.51	8.61

**Table 3 entropy-24-01672-t003:** Efficiency (formula) of the Productive Units of the ORC–VCR system.

Productive Unit	Efficiency		
	Formula		
	E Model	A&F Model	UFS Model
Boiler	$\frac{E_{1:4}}{E_{Q,BLR}}$	$\frac{E_{1:4}^{A}+E_{1:4}^{F}}{E_{Q,BLR}}$	$\frac{E_{1:4}^{U}+E_{1:4}^{F}}{E_{1:4}^{S}+E_{Q,BLR}}$
Expander	$\frac{W_{EXP}}{E_{1:2}}$	$\frac{W_{EXP}+E_{2:1}^{F}}{E_{1:2}^{A}}$	$\frac{W_{EXP}+E_{2:1}^{F}}{E_{1:2}^{U}+E_{2:1}^{S}}$
Condenser	-	$\frac{E_{3:2}^{A}+E_{3:6}^{A}}{E_{6:3}^{F}+E_{2:3}^{F}}$	$\frac{E_{2:3}^{S}+E_{6:3}^{S}}{E_{2:3}^{U}+E_{6:3}^{U}+E_{6:3}^{F}+E_{2:3}^{F}}$
Feed Pump	$\frac{E_{4:3}}{W_{PMP}}$	$\frac{E_{4:3}^{A}+E_{4:3}^{F}}{W_{PMP}}$	$\frac{E_{4:3}^{U}+E_{4:3}^{F}}{E_{4:3}^{S}+W_{PMP}}$
Compressor	$\frac{E_{6:5}}{W_{CMP}}$	$\frac{E_{6:5}^{A}+E_{6:5}^{F}}{W_{CMP}}$	$\frac{E_{6:5}^{U}+E_{6:5}^{F}}{E_{6:5}^{S}+W_{CMP}}$
Evaporator	$\frac{E_{Q,EVP}}{E_{5:7}}$	$\frac{E_{Q,EVP}+E_{5:7}^{F}}{E_{7:5}^{A}}$	$\frac{E_{5:7}^{U}+E_{5:7}^{F}+E_{Q,EVP}}{E_{5:7}^{S}}$
Throttle Valve	-	$\frac{E_{7:3}^{F}}{E_{3:7}^{A}}$	$\frac{E_{7:3}^{F}}{E_{7:3}^{S}+E_{3:7}^{U}}$

**Table 4 entropy-24-01672-t004:** Exegetics unit costs of the productive flows of the Carnot refrigeration cycle by the A&F model.

Flow	Value [kW]	Exergetic Unit Cost [kW/kW]
E^A^_2:1_	13.4	1.00
E^A^_3:2_	16.0	1.00
E^A^_3:4_	4.0	1.00
E^A^_4:1_	25.3	1.00
E^F^_1:4_	13.1	1.00
E^F^_2:1_	0.5	1.00
E^F^_2:3_	16.0	1.00
E^F^_4:3_	2.5	1.00
W_trb_	1.5	1.00
W_cmp_	13.8	1.00
W_cycle_	12.3	1.00
E_Q,evp_	12.3	1.00

**Table 5 entropy-24-01672-t005:** Exergy balances and efficiency (value) of the productive units of the ORC–VCR system.

Productive Unit	E Model				A&F Model				UFS Model			
	Fu (kW)	Pr (kW)	Ir (kW)	η	Fu (kW)	Pr (kW)	Ir (kW)	η	Fu (kW)	Pr (kW)	Ir (kW)	η
Boiler	74.70	51.04	23.66	68.33	74.70	51.04	23.66	68.33	421.90	398.24	23.66	94.39
Expander	40.06	32.43	7.63	80.96	42.09	34.46	7.63	81.87	42.09	34.46	7.63	81.87
Condenser	-	-	-	-	56.17	39.9	16.27	71.03	495.07	478.80	16.27	96.71
Feed Pump	1.52	1.15	0.37	75.86	1.52	1.15	0.37	75.86	1.89	1.52	0.37	80.42
Compressor	20.90	15.86	5.04	75.89	20.90	15.86	5.04	75.89	25.94	20.90	5.04	80.57
Evaporator	9.61	7.55	2.06	78.59	20.20	18.15	2.06	89.82	116.49	114.43	2.06	98.23
Throttle Valve	-	-	-	-	5.02	2.87	2.15	57.17	5.02	2.87	2.15	57.17

**Table 6 entropy-24-01672-t006:** External resource distribution for net power and useful heat.

	W_NET_	Q_EVP_
UFS Model	59.77%	40.23%
A&F Model	50.06%	49.94%

**Table 7 entropy-24-01672-t007:** Exergetic unit cost of the productive flows of the ORC–VCR system by the A&F and UFS models.

Flow	Value [kW]	Exergetic Unit Cost [kW/kW]		Flow	Value [kW]	Exergetic Unit Cost [kW/kW]	
		UFS Model	A&F Model			UFS Model	A&F Model
E^A^_1:2_	42.09	-	3.06	E^F^_4:3_	1.14	5.41	4.93
E^A^_1:4_	12.62	-	1.46	E^F^_5:7_	10.59	3.98	4.94
E^A^_3:2_	29.45	-	3.75	E^F^_6:3_	14.57	4.71	5.5
E^A^_3:6_	10.46	-	3.75	E^F^_6:5_	1.11	5.41	4.93
E^A^_3:7_	5.02	-	4.44	E^F^_7:3_	2.87	7.15	7.75
E^A^_4:3_	0.01	-	4.93	E^S^_1:4_	347.2	3.91	-
E^A^_6:5_	14.76	-	4.93	E^S^_2:1_	7.62	3.91	-
E^A^_7:5_	20.21	-	4.71	E^S^_2:3_	355.2	3.91	-
E^U^_1:2_	34.47	3.6	-	E^S^_4:3_	0.37	3.91	-
E^U^_1:4_	359.8	3.6	-	E^S^_5:7_	116.4	3.91	-
E^U^_2:3_	325.7	3.6	-	E^S^_6:3_	123.6	3.91	-
E^U^_3:7_	2.87	4.23	-	E^S^_6:5_	5.04	3.91	-
E^U^_4:3_	0.38	5.41	-	E^S^_7:3_	2.14	3.91	-
E^U^_5:7_	96.24	3.98	-	W_EXP_	32.42	4.47	3.74
E^U^_6:3_	113.2	4.23	-	W_NET_	10	4.47	3.74
E^U^_6:5_	19.8	5.41	-	W_PMP_	1.52	4.47	3.74
E^F^_1:4_	38.42	3.6	1.46	W_CMP_	20.9	4.47	3.74
E^F^_2:1_	2.04	4.47	3.74	E_Q,EVP_	7.55	3.98	4.94
E^F^_2:3_	41.6	3.69	1.67				

**Table 8 entropy-24-01672-t008:** Modeling complexity analysis.

	A&F	UFS
Flows	22	30
Junctions–bifurcations	4	6
Auxiliary equations	10	16

**Table 9 entropy-24-01672-t009:** Effect of the expander isentropic efficiency on the product-fuel ratio.

Expander Isentropic Efficiency	Product-Fuel Ratio E Model	Product-Fuel Ratio A&F Model	Product-Fuel Ratio UFS Model
0.70	0.71	0.73	0.73
0.73	0.74	0.76	0.76
0.76	0.77	0.78	0.78
0.79	0.80	0.81	0.81
0.82	0.83	0.84	0.84
0.85	0.86	0.86	0.86
0.88	0.89	0.89	0.89
0.91	0.91	0.92	0.92
0.94	0.94	0.95	0.95
0.97	0.97	0.97	0.97
1.00	1.00	1.00	1.00

**Table 10 entropy-24-01672-t010:** Effect of the pump isentropic efficiency on product-fuel ratio.

Pump Isentropic Efficiency	Product-Fuel Ratio E Model	Product-Fuel Ratio A&F Model	Product-Fuel Ratio UFS Model
0.70	0.71	0.71	0.78
0.73	0.74	0.74	0.79
0.76	0.77	0.77	0.81
0.79	0.80	0.80	0.83
0.82	0.83	0.83	0.85
0.85	0.86	0.86	0.87
0.88	0.88	0.88	0.90
0.91	0.91	0.91	0.92
0.94	0.94	0.94	0.95
0.97	0.97	0.97	0.97
1.00	1.00	1.00	1.00

**Table 11 entropy-24-01672-t011:** Effect of the compressor isentropic efficiency on product-fuel ratio.

Compressor Isentropic Efficiency	Product-Fuel Ratio E Model	Product-Fuel Ratio A&F Model	Product-Fuel Ratio UFS Model
0.70	0.71	0.71	0.78
0.73	0.74	0.74	0.79
0.76	0.77	0.77	0.81
0.79	0.80	0.80	0.83
0.82	0.83	0.83	0.85
0.85	0.85	0.85	0.87
0.88	0.88	0.88	0.90
0.91	0.91	0.91	0.92
0.94	0.94	0.94	0.95
0.97	0.97	0.97	0.97
1.00	1.00	1.00	1.00

**Table 12 entropy-24-01672-t012:** Effect of the ΔT_Evaporator_ on product-fuel ratio.

ΔT_Evaporator_	Product-Fuel Ratio E Model	Product-Fuel Ratio A&F Model	Product-Fuel Ratio UFS Model
10	0.78	0.78	0.96
12	0.75	0.75	0.96
14	0.74	0.74	0.96
16	0.72	0.72	0.95
18	0.70	0.70	0.95
20	0.68	0.68	0.94
22	0.67	0.67	0.94
24	0.65	0.65	0.94
26	0.64	0.64	0.93
28	0.63	0.63	0.93

**Table 13 entropy-24-01672-t013:** Effect of the ΔT_boiler_ on product-fuel ratio.

ΔT_Boiler_	Product-Fuel Ratio E Model	Product-Fuel Ratio A&F Model	Product-Fuel Ratio UFS Model
10	0.77	0.77	0.96
12	0.75	0.75	0.96
14	0.73	0.73	0.95
16	0.71	0.71	0.95
18	0.69	0.69	0.94
20	0.68	0.68	0.94
22	0.66	0.66	0.94
24	0.65	0.65	0.93
26	0.63	0.63	0.93
28	0.62	0.62	0.92

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
