# Peer review of "A New Exergy Disaggregation Approach for Complexity Reduction and Dissipative Equipment Isolation in Thermoeconomics"

_entropy, 2022, doi:10.3390/e24111672_

Round 1

Reviewer 1 Report (Previous Reviewer 4)

The authors have responded to all comments and suggestions. Still, I have some major issues with the validity of such an approach. However, it might be the right time to publish it and recieve the comments from readers on the approach and issuers realted.

Best Regards

Author Response

The authors have responded to all comments and suggestions. Still, I have some major issues with the validity of such an approach. However, it might be the right time to publish it and receive the comments from readers on the approach and issuers related.

We would like to inform you that we appreciate your general opinion on our review and also to say thank you for recognizing that this might be the right time to publish.

Also, we believe that future comments from the readers will be useful to improve our research and future works.

Reviewer 2 Report (Previous Reviewer 2)

The authors have made all the considerations of this reviewer. The document is now clearer, the representation of the graphs has been increased and it is easier to understand. Overall, the paper has improved in quality and I believe it will be more visible for future citations.

Author Response

The authors have made all the considerations of this reviewer. The document is now clearer, the representation of the graphs has been increased and it is easier to understand. Overall, the paper has improved in quality and I believe it will be more visible for future citations.

We would like to inform you that we appreciated your general opinion about our revision when you said that “The document is now clearer, the representation of the graphs has been increased and it is easier to understand...” and also for your careful reading about our work.

In addition, we also believe that after the review process was completed, the work will have improved and become more visible for future citations.

Reviewer 3 Report (New Reviewer)

          The work is rather methodological. It considers the energy and exergy balances of a simple power plant. The authors are looking for the most economical way of representing material and energy flows and choose the representation "Helmholtz energy - expansion work". Actually, the article is devoted to the development of such a representation method for the cooling cycle: several approaches are compared, exergy balances are given. To illustrate the operation of the method, a rather simple system has been chosen: it is interesting whether this method will be equally effective for complex plants. Questions and remarks: 1) Thermodynamic functions and variables in the text are not marked in italics, which makes it difficult to read. 2) In formula (8), the value Lo appears, which, apparently, is associated with losses: it should be deciphered.

Author Response

On behalf of all the authors of the article, we would like to thank you for your careful reading of our work. Also, we appreciated your general opinion about our work when you said: “The work is rather methodological.”

Regarding your concern: “To illustrate the operation of the method, a rather simple system has been chosen: it is interesting whether this method will be equally effective for complex plants”

The authors would like to remember that this study is the first in which A&F Model is presented and that complex plants can be analyzed in future work.

Regarding your Questions and remarks:

  • Thermodynamic functions and variables in the text are not marked in italics, which makes it difficult to read.

Response: A careful revision was carried out in order to avoid this mistake and mark in italics all thermodynamic functions and variables in the text.

  • In formula (8), the value Lo appears, which, apparently, is associated with losses: it should be deciphered.

Response: The reviewer was right. Lo is associated with Losses and is now written in the text. Also better explained: Eq. (8) describes a general balance for each unit and the whole plant. The resources (fuel, Fu) expended are not fully converted into the product (Pr) or service due to the irreversibilities (Ir) and losses (Lo) associated with the production process.

This manuscript is a resubmission of an earlier submission. The following is a list of the peer review reports and author responses from that submission.

Round 1

Reviewer 1 Report

Thermo-economics connecting Thermodynamics and Economics concepts has been a hot topic recently for practical and fundamental reasons. It aims to provide information not available in the conventional energy and economic analysis. This work presented a new exergy disaggregation approach, called A&F Model. This model adequately deals with dissipative components, exhibiting less modeling complexity.

The manuscript thus has a certain novelty and might be helpful for engineers. However, I cannot recommend its acceptance in Entropy at least for the present version. The main issue is the error in the citation. Such issues appear everywhere not only for references but also for figures and tables, which make me hard to follow the contents. Also please check the following comments.

1. In general, redundant contents in the template should be removed. For example, text between lines 29-35.

2. Please increase the quality of the figures. Some of them are not clear enough.

3. In Fig. 9, the image and legend overlapped.

I suggest the authors rearrange the manuscript and check it several times to make the draft more readable.

Author Response

Response to Reviewer 1 Comments

Firstly, we would like to inform you that we appreciated your general opinion about our paper when you said, "The manuscript thus has a certain novelty and might be helpful for engineers.” and also for their careful reading of our work. We considered all the modifications you suggested, as explained below:

Point 1: In general, redundant contents in the template should be removed. For example, text between lines 29-35.

Response 1: A careful revision was carried out in order to avoid redundant contents in the template.

Point 2. Please increase the quality of the figures. Some of them are not clear enough.

Response 2: Figures of improved quality have been added to the article.

Point 3. In Fig. 9, the image and legend overlapped.

Response 3: Fig. 9 has been changed to improve quality.

Reviewer 2 Report

The authors propose an A&F model based on exergy disaggregation and apply it to an organic Rankine cycle. The results are compared with the traditional model (E-model) and the UFS model.

The document is difficult to understand mainly because the references of the figures and some tables have failed, probably when transferring it to pdf. In addition to having errors of not deleting part of the template provided by the journal to maintain the format, such as: "Firstname Lastname 1, Firstname Lastname 2 and Firstname Lastname 2,*" or the section "0. How to Use This Template".

In order to improve the document, the following suggestions are proposed:

1.      Check the references of the tables and figures so that the reader does not have to be distracted looking for them.

2.      The authors discuss several models that refer to them by acronyms, considering that not all readers are experts in the field, it would be appreciated if they dedicated part of the introduction to explaining its basic performance, this could make more non-expert readers interested in the document presented, giving it more visibility in the future.

3.      The authors use an accumulation of references to illustrate an idea, it is suggested that authors use no more than three references for these occasions, ideally two. If not, the authors should introduce a sentence explaining why the reference is important for the issue at hand.

4.      The authors make many self-citations to their previous work, of the 46 references approximately 15 are self-citations. I believe that a couple of recent self-citations are sufficient to illustrate the paper presented. Too many self-citations can be interpreted as a lack of up-to-date references to the subject and therefore of interest. It is preferable for a paper to be supported by the work of other researchers and, if possible, by papers that are up to date with the last 4 or 5 years.

5.      As in previous works, the authors could make a comparison with more models, for example in reference 20 they use 4 models of which only two are compared for this case.

6.      In the graphs, the authors use very large markers, which makes some situations where they are very close to each other overlap. If they decrease the size or sometimes draw the line it would be easier to interpret at first view the data shown.

7.      The figures 4 and 5 the subscripts are not clearly readable; they should increase the font size. In general, they should homogenize the font size in all the figures so that they can be differentiated without difficulty in all of them and with the same size.

8.      In their conclusions, the authors reflect on the merits of the proposed model, but state: "It is crucial to say that the A&F model was not proposed to substitute any other thermoeconomic methodology; we therefore all agree that each thermoeconomic model has particular areas of application for which it has proven and solid answers". Please indicate for which particular areas the proposed model is useful.

Author Response

Response to Reviewer 2 Comments

On behalf of all the authors of the article, we would like to thank you for their careful reading of our work. In fact, the references of the figures and some tables have failed, when transferring it to pdf. However, in this last version, these mistakes were resolved.

We considered the majority of modifications you suggested, as explained below:

Point 1. Check the references of the tables and figures so that the reader does not have to be distracted looking for them.

Response 1: All references to the tables and figures were checked in the study.

Point 2. The authors discuss several models that refer to them by acronyms, considering that not all readers are experts in the field, it would be appreciated if they dedicated part of the introduction to explaining its basic performance, this could make more non-expert readers interested in the document presented, giving it more visibility in the future.

Response 2: All the acronyms presented in the introduction are now better explained.

Point 3. The authors use an accumulation of references to illustrate an idea, it is suggested that authors use no more than three references for these occasions, ideally two. If not, the authors should introduce a sentence explaining why the reference is important for the issue at hand.

Response 3: All the references were reviewed and there are no more than three references to illustrate an idea.

Point 4. The authors make many self-citations to their previous work, of the 46 references approximately 15 are self-citations. I believe that a couple of recent self-citations are sufficient to illustrate the paper presented. Too many self-citations can be interpreted as a lack of up-to-date references to the subject and therefore of interest. It is preferable for a paper to be supported by the work of other researchers and, if possible, by papers that are up to date with the last 4 or 5 years.

Response 4: All the references were reviewed and some of them were actualized.

Point 5. As in previous works, the authors could make a comparison with more models, for example in reference 20 they use 4 models of which only two are compared for this case.

Response 5: In fact, the authors have several works in thermoeconomics, and in many of them there are comparisons with more models. However, for this study, the authors chose to compare with two other models, as this is the first work that the A&F Model is presented and also because we chose to compare with the UFS Model, the only model capable of disaggregating the valve. It is worth mentioning, that in the study by d’Accadia & Rossi (1998), a thermoeconomic modeling of a refrigeration system was carried out, disaggregating both the valve and the condenser, and combining the concept of negentropy with the exergy disaggregated into its thermal and mechanical terms. Nevertheless, it maintains the uncommon results associated with the use of negentropy as a fictitious flow; and this type of disaggregation of exergy into thermal and mechanical terms [2] is only advisable, without arbitrariness, for fluids modeled as ideal gas [3]. Additionally, it cannot isolate condensers. In order to achieve consistent values of exergetic unit costs, [4] approached the concept and magnitude of negentropy with a different perspective. This model already considers the dissipative component (condensers and intercoolers) treatment and residue cost allocation naturally through its productive diagram, without the uncommon values obtained when the E&S Model is performed for costs allocation. The H&S Model is considered a consistent approach, but it is not capable of defining a product for any of the types of expansion or bypass valves. The UFS Model [5] was, then, proposed as an extension of the H&S Model to solve this issue. For these reasons, we believe that a comparison with the UFS Model would meet the objectives of this work. Finally, it is worth mentioning that the main aim of this study is present the A&F Model, which is able to isolate dissipative components, including Valves, with less modeling complexity.

Point 6. In the graphs, the authors use very large markers, which makes some situations where they are very close to each other overlap. If they decrease the size or sometimes draw the line it would be easier to interpret at first view the data shown.

Response 6: All graphs presented in the work were reviewed and improved. All markers were replaced.

Point  7. The figures 4 and 5 the subscripts are not clearly readable; they should increase the font size. In general, they should homogenize the font size in all the figures so that they can be differentiated without difficulty in all of them and with the same size.

Response 7: Figures 4 and 5 were reviewed and improved.

Point 8. In their conclusions, the authors reflect on the merits of the proposed model, but state: "It is crucial to say that the A&F model was not proposed to substitute any other thermoeconomics methodology; we therefore all agree that each thermoeconomics model has particular areas of application for which it has proven and solid answers". Please indicate for which particular areas the proposed model is useful.

Response 8: Although the authors have made this comment initially ( It is crucial.... ), after the comment of the reviewer we realize that this phrase can be better written and we decided to delete it.

In the conclusion the authors make clear which particular areas the proposed model is useful and state: The work was proposed to present a new approach for physical exergy disaggregation as a solid and strong option to resolve hard questions in thermoeconomics associated with the treatment and isolation of dissipative equipment, and reduce the complexity in thermoeconomic modeling. And also, this novel approach can be easily applied in different refrigeration systems, especially when there are dissipative components in the cycle under analysis because it is a strong option for disaggregation of each subsystem in the thermodynamic plant.

In future works, the A&F model can be applied to other thermodynamic systems, and based on the results so far, expectations are positive.

References

[1]         d’Accadia MD, Rossi F de. Thermoeconomic optimization of a refrigeration plant. Int J Refrig 1998;21:42–54. https://doi.org/10.1016/S0140-7007(97)00071-6.

[2]         Morosuk T, Tsatsaronis G. Splitting physical exergy: Theory and application. Energy 2019:698–707.

[3]         Lazzaretto A, Tsatsaronis G. SPECO: A systematic and general methodology for calculating efficiencies and costs in thermal systems. Energy 2006;31:1257–89. https://doi.org/10.1016/j.energy.2005.03.011.

[4]         Santos JJCS, Nascimento MAR, Lora EES, Martínez-Reyes AM. On the Negentropy Application in Thermoeconomics: A Fictitious or an Exergy Component Flow? Int J Thermodyn 2009;12:163–76.

[5]         Lourenço AB, Nebra SA, Santos JJCS, Donatelli JLM. Application of an alternative thermoeconomic approach to a two-stage vapour compression refrigeration cascade cycle. J Brazilian Soc Mech Sci Eng 2015;37:903–13. https://doi.org/10.1007/s40430-014-0210-7.

Reviewer 3 Report

One of the problems with thermoeconomic (and exergeconomic) analysis is its complexity and uncertainty. It has been noted by a group of scientists involved in developing of thermoeconomic analysis methods that “while each plant has only one physical structure to describe the physical relations between the process units, various productive structures can be defined depending on the fuel and product definitions …” (Valero et al. “Fundamentals of Exergy Cost Accounting and Thermoeconomics. Part I: Theory”). Therefore, in order to avoid uncertainty the goal of scientists and engineers is to develop simple and accurate thermoeconomic analysis methods that can be applied in real world.

In this study, an attempt to develop new exergy disaggregation method that could help to simplify thermoeconomic analysis has been made. Unfortunately, it seems that this attempt has failed. The proposed approach does not make analysis easier, and it is simply a marginal modification of similar methods that have been presented previously. Also, the method is not clearly presented. For example, it is not clear why the product after compressor is split into to exergy components A and F, and why A component contributes to turbine and evaporator, but F component is used for condenser (Figure 2). Also, strictly speaking, component A (u_i – T_o*s_i) cannot be called Helmholtz energy, because environment temperature T_o is used, while u_i and s_i are measured at temperature T_i (although expression (u_o – T_o*s_o) is Helmholtz energy).

The authors’ attempt to improve Structural Theory of thermoeconomics is commendable. However, the method itself is quite complex and inconvenient compared to other approaches. For example, SPECO method “presents a general, systematic, simple and unambiguous approach for developing the exergetic efficiencies of a thermal system and its components” (Lazzaretto and Tsatsaronis, “SPECO: A systematic and general methodology for calculating efficiencies and costs in thermal systems”) and is easier to implement and apply in real world. Therefore, taking into account, that proposed method only marginally contributes to the development of one specific thermoeconomic analysis method, and the paper fails to provide clear description and advantages of the method, I would not recommend the paper for publication.  

Author Response

Response to Reviewer 3 Comments

On behalf of all the authors of the article, we would like to thank you for the review, also the opportunity to respond to comments.

Responses to the reviewer's comment will be made in parts.

Point 1: The proposed approach does not make analysis easier, and it is
simply a marginal modification of similar methods that have been
presented previously.

Response 1:  The new approach, called the A&F Model, split physical exergy into two terms, namely Helmholtz energy term and flow work term. This is the first time that exergy has been disaggregated in these exergy terms and evaluated from a thermodynamic and thermoeconomic point of view. In our opinion, there is a significant difference between the A&F Model and the other thermoeconomic models. Since the exergy terms used, that is, how exergy was disaggregated is a novelty in thermoeconomics. Also, we believed that A&F Model make analysis easier because it can define their products and/or the fuel for all dissipative components using only two terms of exergy. For example, by comparing the A&F Model and UFS Model represented in a productive diagram for the ORC-VCR system (Figures 4 and 5), it is observed that the A&F model has reduced the quantity of flows by, approximately, 27%, the quantity of junctions-bifurcations by 28.57%, and the quantity of cost equations by 14.29%, in relation to the UFS model. Thus, the complexity reduction in the A&F is significantly smaller than in the UFS Model.

Point 2: Also, the method is not clearly presented.
For example, it is not clear why the product after compressor is
split into to exergy components A and F, and why A component
contributes to turbine and evaporator, but F component is used
for condenser (Figure 2).

Response 2:  The variation of the Helmholtz energy term (EAl:m) and flow work term (EFl:m) components of the exergy between two physical states, l and m, is used to define the flows. The definition of productive flows is attained from the specific exergy terms variation between the inlet and outlet. For example, it is classified as product, when this variation is positive, and the opposite (negative variation) as a fuel[1]. Thus, in Fig. 2 in the compressor, A and F have a positive variation, and consequently are products of the compressor. In the turbine and evaporator, A has a positive variation (product), differently for the condenser which F has a positive variation.

Point 3: Also, strictly speaking, component A (u_i – T_o*s_i) cannot be called Helmholtz energy, because environment temperature T_o is used, while u_i and s_i are measured at temperature T_i (although expression (u_o – T_o*s_o) is Helmholtz energy).

Response 3: 

The concept of the A&F Model begins with the disaggregation proposed by the UFS Model (internal energy term, flow work term and entropic term). It is possible to combine internal energy term and entropic term, such as U - T0S. This rearrangement has a thermodynamic meaning called Helmholtz energy term [3]. The specific Helmholtz energy of a closed system under a heat bath (reservoir at T0) is given by a = u – T0s. This can be applied for both i-th and dead states. Eq.  (1) show the Helmholtz energy term.

                                                                                 (1)

It is worth mentioning that is a term of exergy (a component). We have to pay attention that the  (Helmholtz energy term) is “ai - a0”.

Point 4:  However, the method itself
is quite complex and inconvenient compared to other
approaches. For example, SPECO method “presents a general,
systematic, simple and unambiguous approach for developing
the exergetic efficiencies of a thermal system and its
components” (Lazzaretto and Tsatsaronis, “SPECO: A
systematic and general methodology for calculating efficiencies
and costs in thermal systems”) and is easier to implement and
apply in real world.

Response 4:  Although the authors recognize that Speco is a reference in exergoeconomics, this approach presents a limitation with dissipative component treatment (the isolation of condensers, valves, intercoolers, etc.), which lies in the impossibility of defining its products and/or fuels using only the total exergy flow. For dissipative components, a meaningful exergetic efficiency cannot be defined [4], and All costs associated with a dissipative component should be appropriately apportioned among all other components served by it [4].  Bearing this in mind, this study presents this new physical exergy disaggregation approach, A&F Model, which adequately deals with dissipative components (condenser, intercoolers, and valve) with less modeling complexity. Finally, it is worth mentioning, that considering separate exergy forms improves the accuracy of the results [4].

References

[1]         Santos JJCS, Nascimento MAR, Lora EES, Martínez-Reyes AM. On the Negentropy Application in Thermoeconomics: A Fictitious or an Exergy Component Flow? Int J Thermodyn 2009;12:163–76.

[2]         Erlach B, Serra L, Valero A. Structural theory as standard for thermoeconomics. Energy Convers Manag 1999;40:1627–49. https://doi.org/10.1016/S0196-8904(99)00057-6.

[3]         Moran MJ, Shapiro HN, Boettner DD, Bailey MB. Fundamentals of Engineering Thermodynamics. 7th ed. John Wiley and Sons; 2011.

[4]         Lazzaretto A, Tsatsaronis G. SPECO: A systematic and general methodology for calculating efficiencies and costs in thermal systems. Energy 2006;31:1257–89. https://doi.org/10.1016/j.energy.2005.03.011.

Reviewer 4 Report

Dear Authors,

Recently, I had the opportunity to review the submitted manuscript "A New Exergy Disaggregation Approach for Complexity Reduction and Dissipative Equipment Isolation in Thermoeconomics" that proposes a new method for disaggregation of exergy flows in order to reduce the complexity seen in thermoeconomics (i.e. exergioeconomics). 

I believe exergy analysis is crucial for better understanding of thermal systems and help in obtaining insights into such systems allowing the planners/engineers and designers to optimize thermal systems with the exergioeconomics tool. Therefore, the complexity reduction in their methods are essential. However, such a reduction should not be carried out on the cost of accuracy or principles of thermodynamics. Throughout the review process, I collected the following:

1-      The introduction section should be significantly improved and further research studies should be included in the literature review in order to highlight the current gap and outline the work’s novelty. Otherwise, it might be challenging to see any contribution to the main body of knowledge.

2-      Literature review must be expanded.

3-      Methodology section should be better explained in order to allow other scholars to replicate the methods adopted.

4-      All abbreviations used in the work should be introduced within the text and explained.

5-      Considering table 5: the results reported in this table highlight the incapability (weakness) of this proposed method as it shows inaccuracy for heat exchanging units (i.e. condenser, boiler, evaporator) whereas it reports similar results for mechanical units (e.g. fan). I believe this might be attributed to exclusion of latent heat in such units. Perhaps the authors can check their models and double check the results whether their proposed analysis method considers latent heat. Besides, it is not fundamentally clear why there is a discrepancy between A&F and UFS. If both models are developed using same thermodynamic principles, shouldn’t we expect similar/close results? This should be further elaborated.

6-      Many sections over the entire manuscript show several errors when referring to tables and figures. Please revise this.

7-      The authors should further elaborate on the adoption of Carnot refrigeration cycle as the main case study in this work. The work should also consider new technologies and test the applicability of such a method for these technologies. Such technologies can be PV panels, thermal collectors, seasonal heat storage, hydrogen storage, electrolyzes and fuel cells. Besides, it would be interesting to test this method for chemical processes whereby reaction kinetics are expected. Such a process could be drying.

8-      The figures should be improved.

9-      The manuscript has several language flaws and typos. Please consider a proofread by a native speaker to improve the language used.

10-  Please double check section (0. How to Use This Template) as this is not needed in the manuscript.

Best Regards

Author Response

Response to Reviewer 4 Comments

On behalf of all the authors of the article, we would like to thank you for their careful reading of our work. We considered the majority of modifications you suggested, as explained below:

Point 1- The introduction section should be significantly improved and further research studies should be
included in the literature review in order to highlight the current gap and outline the work’s novelty. Otherwise, it might be challenging to see any contribution to the main body of knowledge.

Response 1: The introduction has been revised, expanded, and improved.

Point 2- Literature review must be expanded.

Response 2: The Literature review has been expanded.

Point 3- Methodology section should be better explained in order to allow other scholars to replicate the methods adopted.

Response 3: The methodology section was reread and a few modifications were made. Anyway, if the reviewer still feels that further improvement is needed. We would kindly ask you to be more specific as to what needs to be further explained.

Point 4- All abbreviations used in the work should be introduced within the text and explained.

Response 4: All the abbreviations presented in the manuscript were revised and are now better explained.

Point 5- Considering table 5: the results reported in this table highlight the incapability (weakness) of this proposed method as it shows inaccuracy for heat exchanging units (i.e. condenser, boiler, evaporator) whereas it reports similar results for mechanical units (e.g. fan). I believe this might be attributed to exclusion of latent heat in such units. Perhaps the authors can check their  models and double check the results whether their proposed analysis method considers latent heat. Besides, it is not fundamentally clear why there is a discrepancy between A&F and UFS. If both models are developed using same thermodynamic principles, shouldn’t we expect similar/close results? This should be further elaborated.

Response 5: Table 5 shows the Efficiency (value) of the Productive Units of the ORC-VCR system. By using the A&F Model, the product-fuel ratios (efficiency) has its results between zero (if the processes are totally irreversible) and 1 (if the processes are totally reversible) for each component present in the plant (even the dissipative ones, such as the condenser), confirming been coherent of thermodynamic point of view. Moran et al. (2011) stated that efficiency gauges how effectively the input is converted to the product. Hence, the concepts of fuel (Fu), product (Pr), irreversibility (Ir), and efficiency (ƞ) are not independent in thermoeconomics. In other words, the efficiency defined by the product fuel ratio is, quantitatively, the parameter for costs generation index. In the A&F Model the physical exergy is disaggregated into Helmholtz energy term and flow work term. The products and the fuels of all subsystems, in terms of Helmholtz energy term and flow work term, are determined in accordance with the quantity of these magnitudes added to and removed from the working fluid, respectively. It is the same principle utilized for the UFS Model, However, the products and fuel are determined by internal energy, flow work, and entropic terms. Thus, despite using the same thermodynamic principles, the results can be different once the terms of exergy utilizes are different. Finally, Depending on the productive structure definition, different cost values can be obtained (Valero et al. 2006).

Point 6- Many sections over the entire manuscript show several errors when referring to tables and figures. Please revise this.

Response 6: All references to the tables and figures were checked in the study.

Point 7- The authors should further elaborate on the adoption of Carnot refrigeration cycle as the main case study in this work. The work should also consider new technologies and test the applicability of such a method for these technologies. Such technologies can be PV panels, thermal collectors, seasonal heat storage, hydrogen storage, electrolyzes and fuel cells. Besides, it would be interesting to test this method for chemical processes whereby reaction kinetics are expected. Such a process could be drying.

Response 7: In the study, The Carnot Refrigeration Cycle and a system combining the Organic Rankine Cycle (ORC) and Vapor Compression Refrigeration (VCR) cycle (ORC-VCR system), are evaluated from a thermodynamic and thermoeconomic point of view through cost allocation using the A&F Model. However, in the study, the authors make clear that the Carnot refrigeration cycle (a simple example) is used to assess the consistency and coherence of the model both from a thermodynamic and thermoeconomic point of view. It is worth mentioning that the main case study in the work is ORC-VCR. The authors chose the ORC-VCR system to compare the A&F model with the UFS model and performed a design point and parametric studies. Finally, the authors would like to say thank you for the suggestions for considering new technologies and test the applicability of the method, and answer that these news technologies can be performed in future work, and remember that this study is the first which A&F Model is presented.

Point 8- The figures should be improved.

Response 8: All figures presented in the work were reviewed and improved.

Point 9- The manuscript has several language flaws and typos. Please consider a proofread by a native speaker to improve the language used.

Response 9: A careful revision was carried out in order to avoid or minimize grammar errors and also to adjust the language.

Point 10- Please double check section (0. How to Use This Template) as this is not needed in the manuscript.

Response 10: There is no more section 0 in the manuscript.

References

Moran, M. J., Shapiro, H. N., Boettner, D. D., & Bailey, M. B. (2011). Fundamentals of Engineering Thermodynamics, 7th edn, John Wiley and Sons.

Valero, A., Serra, L., & Uche, J. (2006). Fundamentals of Exergy Cost Accounting and Thermoeconomics. Part I: Theory. Journal of Energy Resources Technology, 128(1), 1–8.

Round 2

Reviewer 1 Report

The authors answered all my questions and the draft is much improved. I recommend its acceptance.

Reviewer 2 Report

The authors have made an effort to address most of the considerations raised, although this issue raised in the previous review has not been adequately addressed:

"Point 4. The authors make many self-citations to their previous work, of the 46 references approximately 15 are self-citations. I believe that a couple of recent self-citations are sufficient to illustrate the work presented. Too many self-citations can be interpreted as a lack of up-to-date references on the subject and therefore of interest. It is preferable for a paper to rely on the work of other researchers and, if possible, on up-to-date work from the last 4 or 5 years". 

In the new paper there are 13 self-citations, I insist again that no more than a couple of self-citations should be left out, not a couple reduced.

Reviewer 3 Report

The manuscript has not been sufficiently improved. Therefore, as I have mentioned in my previous comments I would not recommend accepting the paper for publication.

Reviewer 4 Report

Dear Authors,

I still believe that if both methods emerge from same thermodynamic principles, they should necessarily lead to the same results. Otherwise, there is a fault either in fundamentals or interpretation of fundamentals. 

Best Regards